# Associations of multiple trace elements with bipolar disorder in adolescents: A case-control study

Jie Li[1,2☉], Xuemei Li[1,2☉], Yuqian He[1,2], Yajie Huang[1,2], Wenjing Wang[1,2], Hang Du[3], Chengzhi Chen[4], Dan Zhu[5‡*], Xinyu Zhou[1,2‡*]

1 Department of Psychiatry, The First Affiliated Hospital of Chongqing Medical University, Chongqing, China, 2 Key Laboratory of Major Brain Disease and Aging Research (Ministry of Education), Chongqing Medical University, Chongqing, China, 3 Chongqing prevention and treatment center for occupational diseases, The First Affiliated Hospital of Chongqing Medical and Pharmaceutical College, Chongqing, China, 4 Department of Occupational and Environmental Health, School of Public Health, Chongqing Medical University, Chongqing, China, 5 Department of Neurology, The First Affiliated Hospital of Chongqing Medical University, Chongqing, China

☉ These authors contributed equally to this work.
‡ These authors jointly supervised this work.
* zhouxinyu@cqmu.edu.cn (XZ); danzhu25@163.com (DZ)

## Abstract

### Background

Bipolar disorder (BD) is a serious mental disorder. Studies have shown an association between trace elements and mental disorders. However, this association has not been thoroughly studied in adolescents with BD. We aimed to investigated the associations between multiple trace elements and adolescent BD.

### Method

This case-control study included 144 BD patients with BD and 144 matched controls. Seventeen elements in the participants' urine were measured using inductively coupled plasma mass spectrometry (ICP-MS). Least absolute shrinkage and selection operator (LASSO), multivariate logistic regression, restricted cubic spline (RCS), and Bayesian kernel machine regression (BKMR) were used to analyze the association between exposure to single and mixed elements and adolescent BD.

### Results

In the single-element models, titanium, manganese, rubidium, and iodine were negatively associated with adolescent BD. In the multi-element model selected by LASSO, titanium (OR = 0.14, 95% CI: 0.04–0.53), manganese (OR = 0.02, 95% CI: 0.01–0.08), and iodine (OR = 0.06, 95% CI: 0.02–0.22) showed a negative correlation with adolescent BD, while magnesium (OR = 11.24, 95% CI: 1.83–69.12), and nickel

**Data availability statement:** All relevant data are available in the Figshare repository (https://doi.org/10.6084/m9.figshare.28332461.v1)

**Funding:** This work was supported by the Major Program of Brain Science and Brain-Like Intelligence Technology (2022ZD0212900 to XZ), the National Natural Science Foundation of China (82271565 to XZ), the Natural Science Foundation of Chongqing (CSTB2023NSCQ-MSX0381 to DZ). The funders had no role in study design, data collection and analysis, decision to publish, or preparation of the manuscript

(OR = 6.86, 95% CI: 1.55–30.29) displayed a positive correlation. The RCS results showed a non-linear correlation between the elements titanium, manganese, iodine, magnesium, nickel, zinc, strontium and adolescent BD. In addition, the BKMR analysis showed a significant joint effect of multiple elements on adolescent BD when the concentrations of the seven elements were at or above the 55th percentile, compared with their median values.

## Conclusions

Our findings revealed that urinary titanium, manganese, and iodine were negatively correlated with adolescent BD, whereas urinary magnesium and nickel were positively correlated with adolescent BD. These results provide evidence of an association between urinary trace elements and adolescent BD.

## Introduction

Bipolar disorder (BD), a serious mental disorder, frequently emerges during adolescence [1,2]. Of adults with BD, 15–28% experience illness onset before the age of 13 years and 50–66% before the age of 19 years [3,4]. BD is characterized by mood instability with alternating periods of depression, mania or hypomania, and euthymic states, which leads to severe impairment of cognitive and social function in patients and imposes an enormous disease burden on families and society [5]. However, the pathogenesis of BD has yet to be elucidated.

The traditional risk factors, including psychological stress, genetic, and biochemical factors, can only partially explain the causes of BD [6,7]. In recent years, accumulating epidemiological studies have evaluated the potential association between levels of various elements from different biological samples and the prevalence of BD, but the results are inconsistent [8,9]. Studies have reported that the serum concentration of magnesium (Mg) is positively associated with the prevalence of BD [10]. Nevertheless, another study observed an inverse correlation between serum Mg levels and BD risk [11]. Previous studies have found that vanadium (V), an essential element, is a causative factor for BD, and elevation of V and molybdenum (Mo) has been reported in serum samples of patients with BD [12,13]. In addition, recent studies have confirmed that the concentration of copper (Cu) in the affected group was lower than that in the control group, however for the zinc (Zn) concentrations, an inverse relationship was observed in Polish adults [9].

Elements commonly coexist in almost all environmental media, including drinking water, ambient air, food, and humans are often simultaneously exposed to multiple elements [14,15]. However, the relationship between multiple elemental levels and adolescent BD remains unclear. Inter-element interactions may alter the toxicity of single elements. Therefore, the co-exposure effect of multiple elements cannot be ignored in the study of elements. Studies have reported that traditional multivariate regression models (adjusted for other elements) could be biased or insufficient to estimate the effects of elements on health [16]. Bayesian kernel machine regression

(BKMR) can evaluate the joint effect of the components of a mixture, allowing for potential non-linear effects and interactions [17,18]. Thus, the BKMR model should be used to explore the combined t effects of multiple factors on adolescent BD.

We conducted a case-control study in Southwest China, and measured the urinary levels of 17 elements. Using various statistical methods, including least absolute shrinkage and selection operator (LASSO), multivariate logistic regression, restricted cubic spline (RCS), and BKMR, the combined effects of various trace elements were investigated. We also analyzed the interaction between these factors and the effect of mixed concentrations on adolescent BD.

## Materials and methods

### Ethics statement

This study was conducted in accordance with the principles of the Declaration of Helsinki. This study was approved by the Medical Ethics Committee of the First Affiliated Hospital of Chongqing Medical University (Ethical approval number: 2020-864). All participants and their legal guardians signed informed consent forms.

### Study population

Recruitment and data collection were conducted between May 2023 and January 2024. A total of 144 adolescents with BD were recruited from the Department of Psychiatry at the First Affiliated Hospital of Chongqing Medical University. All participants were carefully screened by two trained senior psychiatrists to confirm that they met the Diagnostic and Statistical Manual of Mental Disorders, 5th edition (DSM-5) criteria for BD. The controls were recruited through advertisements at the same hospital and were matched to each case (1:1) according to gender and age (± 2 years).

Inclusion criteria for all participants were as follows: (1) age ≤18 years, (2) meeting the diagnostic criteria for BD according to the DSM-5. The exclusion criteria were as follows: (1) presence or history of severe medical, neurological, or psychiatric disorders, (2) substance abuse, and (3) currently menstruating.

### Questionnaire and variable definition

In the survey, trained interviewers used a structured questionnaire to collect information about demographic characteristics, including gender, age, body mass index (BMI), residence, monthly household income, physical activity, sleep, smoking, alcohol consumption, family history of mental diseases, suicidal behavior, and non-suicidal self-injury (NSSI). BMI was calculated as weight divided by height squared and categorized into underweight (<18.5 kg/m²), normal weight (18.5–23.9 kg/m²), overweight (24.0–27.9 kg/m²) and obese (≥28.0 kg/m²) [19]. Participants reported their average monthly household income using classifications consistent with the China Statistical Yearbook (published in 2020): <3000 RMB, 3000–6000 RMB, 6000–12000 RMB, and >12000 RMB. Hours of leisure-time physical activity per week were divided into five groups: <1 h, 1–3 h, 3–5 h, 5–7 h, and >7 h. Daily sleep time was categorized into two groups: < 8 h and ≥ 8 h. In the current study, smokers were defined as those who smoked at least one cigarette per day for more than six months. Drinkers were defined as those who drank at least once a month on average for more than one year.

Depression and anxiety symptom severity were assessed using the 17-item Hamilton Depression Scale (HAMD-17) [20] and the 14-item Hamilton Anxiety Rating Scale (HAMA-14) [21]. The Young Mania Rating Scale (YMRS) [22] was used to assess symptoms of mania. In addition, patients who were currently depressed, and had HAMD-17 score ≥ 17 and YMRS score < 12, were classified as bipolar depression patients who were currently manic, and those with HAMD-17 score < 17 and YMRS score ≥ 12 were classified as bipolar manic/hypomania patients [23].

### Measurement of urinary elements

In this study, urine samples were collected and stored immediately at -80°C until analysis. Element concentrations were determined using inductively coupled plasma mass spectrometry (ICP-MS, Perkin Elmer NexION 300X, Shelton, CT,

USA). Frozen urine samples were thawed at room temperature, and 0.2 mL from each sample was then diluted with 3.8 mL of 1% (v/v) nitric acid. After digestion at 90°C and centrifugation, the supernatant was extracted for automatic sampling. Seronorm Trace Elements Urine L-2 (SERO AS, Billingstad, Norway) was used for quality control to ensure accuracy of the laboratory measurements. The urinary element levels were corrected using urinary creatinine concentration, which was measured using the sarcosine oxidase approach with a BECKMAN DXC 800 automatic biochemistry analyzer. Measurements below the limit of detection (LOD) were replaced with LOD/$\sqrt{2}$ [24]. Notably, the detection rates of all elements were >70%, which relied on the LODs (Table S1 in S1 File).

## Statistical analysis

The demographic characteristics of the study participants were descriptively analyzed. According to the skewness of the data, Wilcoxon signed-rank tests were used for continuous variables, and chi-squared tests were used for categorical variables. Spearman correlation was used to calculate correlations between elements. Element levels were corrected for creatinine before analysis and then transformed by natural logarithms. Medians and percentiles were calculated to describe the distribution of the elements in both groups. We used multivariate logistic regression models to evaluate the odds ratios (ORs) and 95% confidence intervals (CIs) of adolescent BD. The levels of each element were categorized into quartiles, according to the concentration distribution. The lowest quartile was designated as the reference.

We built two models to analyze the results for a single element. Model 1 was adjusted for gender and age, and Model 2 was additionally adjusted for other demographic characteristics as covariates, including BMI, residence, monthly household income, physical activity, sleep, smoking, alcohol consumption, and family history of mental diseases. Linear trend p-values were calculated by entering the concentration of each element into the models using the lowest quartile as the reference. Furthermore, RCS regression models were applied to analyze the dose-response associations between element concentrations and adolescent BD.

LASSO is a penalized regression method that shrinks the absolute value of the regression coefficients, potentially to zero, and therefore retains the variables most relevant to the outcome [25]. To enhance the predictive accuracy and interpretability of the model, we employed LASSO regression to select representative elements for the multi-element model. LASSO produces a more refined model by constructing a penalty function and sets some regression coefficients to zero. Thus, it can retain the advantage of subset shrinkage, thereby avoiding overfitting [26,27].

To accurately reflect the nonlinear effect of elements on BD, we applied BKMR to assess the overall associations of exposure to multiple elements with adolescent BD and to investigate the possible interactions and exposure-responses of the relevant elements. The BKMR function is [17,18]:

$$Y_i = h \ (Mg, Ti, Mn, Ni, Zn, Sr, I) \ +\beta X_i + \varepsilon_i$$

where $Y_i$ represents the response for each adolescent i (i = 1,…,n), h is the exposure–response function among the seven elements, and $X_i$ and β represent covariates and their coefficients, respectively.

All statistical analyses were performed using IBM SPSS (version 26.0, Armonk, NY, USA) and R software (v.4.1.0), and a two-tailed p < 0.05 was considered statistically significant.

## Results

### Characteristics of participants

This study comprised 144 patients diagnosed with BD and 144 healthy controls. Females comprised the majority (73.6%). Significant differences were observed between the patients and control participants in terms of physical activity (p = 0.009) and suicidal behavior (p < 0.001). In the case group, 67.4% of the patients exhibited NSSI behavior, with the majority currently experiencing depressed mood (61.8%). Additionally, 96.6% of the patients were receiving treatment. The YMRS

(p < 0.001), HAMD-17 (p < 0.001), and HAMA-14 (p < 0.001) scores were significantly higher in the case group than in the control group. However, there were no significant differences in BMI, residence, monthly household income, sleep, smoking, alcohol consumption, family history of mental diseases, or urinary creatinine levels (all p > 0.05) (Table 1).

**Urinary levels of trace elements**

We compared the level of each element (adjusted for urinary creatinine) in the cases and controls using the Wilcoxon signed-rank test and summarized the distribution of the regulated concentrations of the 17 elements. As shown in Table S2 in S1 File, the levels of V and As of the cases were generally higher than controls, in contrast, the concentrations of Ti, Mn, Rb, Te, I, B, Mg, Al, Fe, Co, Ni, Cu, Zn, Sr, and Mo were lower than controls. The differences were significantly smaller for Ti, Mn, Rb, Te, and I (p < 0.05). Spearman correlations are shown in Fig S1 in S1 File.

**Urinary trace elements and adolescent BD**

The logistic regression analysis presented in Table 2 illustrates the association between urinary trace elements and BD. We categorized the concentrations of each element, corrected for creatinine and transformed by natural logarithm, into quartiles. The lowest concentration group (quartile 1) was used as the reference in the model. In Model 1, Ti, Mn, Rb, and I exhibited significant negative correlations with adolescent BD, which persisted after adjusting for demographic factors in Model 2 (p < 0.05). In the highest quartiles, negative correlations with BD were observed for concentrations of Ti, Mn, Rb, and I (quartile 4 vs. quartile 1: OR = 0.21, 95% CI: 0.09–0.49; OR = 0.05, 95% CI: 0.02–0.15; OR = 0.15, 95% CI: 0.06–0.37; OR = 0.10, 95% CI: 0.04–0.25, respectively). Boron, initially non-significant in Model 1, showed a correlation after adjusting for population factors in Model 2 (p = 0.046). There were no statistically significant differences between the other elements and BD (p > 0.05).

**Multiple-element model for adolescent BD**

To validate the rationale of our multi-element model, we further refined these elements using LASSO regression, and identified a multi-element model that included Mg, Ti, Mn, Ni, Zn, Sr, and I (Fig 1). By multivariate logistic regression, these seven trace elements were collectively included and analyzed along with demographic factors (Table 3). The results showed that Ti, Mn, and I were negatively correlated with adolescent BD, with the highest quartile (quartile 4) having ORs of 0.14 (95% CI: 0.04–0.53), 0.02 (95% CI: 0.01–0.08) and 0.06 (95% CI: 0.02–0.22), respectively. Notably, Mn and I showed inhibitory effects on BD, regardless of the concentration levels at which the OR values were <1 compared to their minimum quartiles. Conversely, increasing Mg, Ni, Zn, and Sr quartiles remained positively related with adolescent BD, with the ORs in highest quartile being 11.24 (95% CI:1.83–69.12), 6.86 (95% CI:1.55–30.29), 2.22 (95% CI:0.58–8.55), and 2.18 (95% CI:0.49–9.76), respectively.

Adjusted for gender, age, BMI, residence, monthly household income, physical activity, sleep, smoking, alcohol consumption, family history of mental diseases and seven elements. Mg: magnesium, Ti: titanium, Mn: manganese, Ni: nickel, Zn: zinc, Sr: strontium, I: iodine.

**Dose-response relationships of element level and adolescent BD**

The dose-response associations between elements and adolescent BD were assessed using RCS based on the results of the multiple-element logistic regression model and LASSO regression, which included Mg, Ti, Mn, Ni, Zn, Sr, and I. For urinary Mn, Ti, and I levels, RCS analyses showed negative associations with BD (Mn: overall association p < 0.001, p for non-linearity < 0.001; Ti: overall association p < 0.001, p for non-linearity = 0.004; I: overall association p < 0.001, p for non-linearity = 0.375). A linear dose-response relationship showed a positive correlation between urinary Zn, Mg, Ni, Sr, and BD (Zn: overall association p < 0.001, p for non-linearity < 0.001; Mg: overall association p < 0.001, p for

**Table 1. Baseline characteristics of cases and controls.**

| Characteristics | Case (n = 144) | Control (n = 144) | p value |
|---|---|---|---|
| Girls, n (%) | 106 (73.60) | 106 (73.60) | 1.000 |
| Age, years, mean (SD) | 15.27 ± 0.14 | 15.10 ± 0.14 | 0.580 |
| BMI, kg/m², n (%) | 144 | 144 | 0.213 |
| <18.5 | 41 (28.40) | 47 (32.60) | |
| 18.5-23.9 | 76 (52.80) | 80 (55.60) | |
| 24.0-27.9 | 21 (14.60) | 10 (6.90) | |
| ≥28.0 | 6 (4.20) | 7 (4.90) | |
| Resident, n (%) | 144 | 144 | 0.235 |
| urban | 137 (95.10) | 132 (91.70) | |
| rural | 7 (4.90) | 12 (8.30) | |
| Monthly household income, RMB, n (%) | 144 | 144 | 0.454 |
| ≤3000 | 23 (16.00) | 15 (10.40) | |
| 3000-6000 | 71 (49.30) | 69 (47.90) | |
| 6000-12000 | 35 (24.30) | 42 (29.20) | |
| >12000 | 15 (10.40) | 18 (12.50) | |
| Leisure time physical activity per week, h, n (%) | 144 | 144 | 0.009 |
| <1 | 36 (25.00) | 25 (17.40) | |
| 1-3 | 46 (31.90) | 70 (48.60) | |
| 3-5 | 29 (20.20) | 33 (22.90) | |
| 5-7 | 18 (12.50) | 7 (4.80) | |
| ≥7 | 15 (10.40) | 9 (6.30) | |
| Daily sleep time, h, n (%) | 144 | 144 | 0.802 |
| <8 | 96 (66.70) | 98 (68.10) | |
| ≥8 | 48 (33.30) | 46 (31.90) | |
| Smoking status, n (%) | 144 | 144 | 0.643 |
| No | 133 (92.40) | 135 (93.80) | |
| Yes | 11 (7.60) | 9 (6.20) | |
| Alcohol consumption, n (%) | 144 | 144 | 0.312 |
| No | 133 (92.40) | 128 (88.90) | |
| Yes | 11 (7.60) | 16 (11.10) | |
| Family history of mental diseases, n (%) | 144 | 144 | 0.090 |
| No | 137 (95.10) | 142 (98.60) | |
| Yes | 7 (4.90) | 2 (1.40) | |
| Suicide, n (%) | 144 | 144 | <0.001 |
| No suicidal intent | 22 (15.30) | 139 (97.20) | |
| Suicidal intent | 39 (27.10) | 3 (2.10) | |
| Suicidal attempt | 23 (16.00) | 0 (0.00) | |
| Suicide attempt survivor | 60 (41.70) | 2 (1.40) | |
| NSSI, n (%) | 144 | 144 | <0.001 |
| No | 47 (32.60) | 141 (97.90) | |
| Yes | 97 (67.40) | 3 (2.10) | |
| Current mood, n (%) | 144 | | |
| Manic | 55 (38.20) | | |
| Depression | 89 (61.80) | | |
| Treatment, n (%) | 144 | | |

*(Continued)*

**Table 1.** (Continued)

| Characteristics | Case (n = 144) | Control (n = 144) | p value |
|---|---|---|---|
| No | 5 (3.4) | | |
| Yes | 139 (96.6) | | |
| YMRS, mean (SD) | 10.87 ± 0.38 | 1.90 ± 0.14 | <0.001 |
| HAMD-17, mean (SD) | 17.17 ± 0.49 | 1.75 ± 0.17 | <0.001 |
| HAMA-14, mean (SD) | 11.23 ± 0.58 | 1.58 ± 0.14 | <0.001 |
| Urine creatinine, mean (SD) | 1.75 ± 0.08 | 1.59 ± 0.06 | 0.298 |

BMI: body mass index, YMRS: Young Mania Rating Scale, HAMD-17: 17-item Hamilton Depression Rating Scale, HAMA-14: 14-item Hamilton Anxiety Rating Scale, NSSI: non-suicidal self-injury, SD: standard deviation.

non-linearity < 0.001; Ni: overall association p = 0.017, p for non-linearity = 0.009; Sr: overall association p = 0.01, p for non-linearity = 0.005) (Fig 2).

## BKMR analyses

Based on the results of LASSO regression, we included "representative" elements in the BKMR analysis. We simulated individual exposure-response functions for each element, fixing the concentrations of other elements at their median levels (95% CI). In the univariate cross-sectional diagram (Fig 3a), with an increase in Mg, Sr, and Ni concentrations, the positive correlation with BD showed an increasing trend. The Zn effect curve did not show any evident increase or decrease. Mn showed a change that first decreased and then increased, and the change curve was asymmetric. In addition, the effect curves of Ti and I first showed a slow rise followed by a rapid decline.

Fig 3b shows the overall correlations of the mixed elements. In these cases, the joint effect was statistically significant when all elements were compared to the median, and they were all found to have a positive impact on the outcome.

An alternative method of describing the effects of exposure involves assessing the distribution of individual exposures in relation to their cumulative impact. For instance, the correlation differs when a single exposure is at the 75th percentile versus the 25th percentile, with all other elements fixed at specific quantiles, such as P25, P50, and P75. In Fig 3c, a negative correlation between Mn and I exposure and adolescent BD (75th vs 25th percentiles) was investigated, as the other elements were fixed at disparate percentiles (25th, 50th, or 75th). Regardless of the concentrations of the other elements, P75 Mn had a lower effect than P25 Mn. When the other elements were fixed at the P25 and P50 percentiles, although the Ti concentration increased to the 75th percentile, the effect continued to increase; however, it showed no statistical significance, approaching zero. In contrast, the negative effects of I increased with increasing concentrations. In addition, the effects of Mg, Ni, Zn, and Sr at different concentrations were not statistically significant, regardless of the concentrations of the other elements.

We continued to explore the bivariate exposure-response consequences as shown Fig 3d. Each row represents "exposure 2", located at P25, P50, and P75. Each column is the studied element ("exposure 1"), and others are fixed at their medians. These curves indicate whether an interaction exists between the elements. If the curves are largely parallel, there is no significant interaction, however, if they diverge or intersect, an interaction may occur. We observed a potential joint effect of Zn, Mn, and I, and an interaction between Ni and Mn.

## Subgroup analyses

In this study, manic patients showed significantly higher levels of urinary As than depressed patients (Table S3 in S1 File). In the single-element models, the urinary As ORs was 4.56 (95% CI:1.31–15.89), which remained statistically significant after adjusting for other covariates in Model 2 (Table S4 in S1 File). No obvious representative elements were identified in the LASSO analysis.

**Table 2. Odds ratios (ORs) and 95% confidence intervals (CIs) for adolescent BD based on urinary elements in the single-element model.**

| Elements | Quartile 1 | Quartile 2 | Quartile 3 | Quartile 4 | p trend |
|---|---|---|---|---|---|
| B | | | | | |
| Range | ≤5.78 | ~6.33 | ~6.86 | ≥6.87 | |
| Case/Control | 27/36 | 53/36 | 52/36 | 12/36 | |
| Model 1 | 1.00 (ref) | 1.96 (1.01-3.78) | 1.90 (0.97-3.71) | 0.44 (0.19-1.01) | 0.122 |
| Model 2 | 1.00 (ref) | 1.75 (0.86-3.54) | 1.55 (0.76-3.14) | 0.36 (0.15-0.86) | 0.046 |
| Mg | | | | | |
| Range | ≤10.17 | ~10.85 | ~11.29 | ≥11.30 | |
| Case/Control | 18/36 | 46/36 | 47/36 | 33/36 | |
| Model 1 | 1.00 (ref) | 2.55 (1.25-.5.21) | 2.61 (1.27-5.35) | 1.81 (0.86-3.79) | 0.160 |
| Model 2 | 1.00 (ref) | 2.47 (1.13-5.37) | 2.34 (1.07-5.09) | 1.64 (0.73-3.66) | 0.301 |
| Al | | | | | |
| Range | ≤-.16 | ~1.30 | ~2.31 | ≥2.32 | |
| Case/Control | 27/36 | 52/36 | 39/36 | 26/36 | |
| Model 1 | 1.00 (ref) | 1.945 (1.01-3.76) | 1.45 (0.74-2.86) | 0.97 (0.47-1.98) | 0.950 |
| Model 2 | 1.00 (ref) | 2.59 (1.25-5.36) | 1.79 (0.84-3.79) | 1.12 (0.50-2.47) | 0.791 |
| Ti | | | | | |
| Range | ≤6.92 | ~7.48 | ~8.00 | ≥8.01 | |
| Case/Control | 46/36 | 52/36 | 35/36 | 11/36 | |
| Model 1 | 1.00 (ref) | 1.13 (0.61-2.07) | 0.75 (0.40-1.43) | 0.24 (0.11-0.53) | 0.001 |
| Model 2 | 1.00 (ref) | 1.33 (0.68-2.59) | 0.70 (0.35-1.42) | 0.21 (0.09-0.49) | 0.001 |
| V | | | | | |
| Range | ≤2.75 | ~3.27 | ~3.76 | ≥3.77 | |
| Case/Control | 32/36 | 36/36 | 33/36 | 43/36 | |
| Model 1 | 1.00 (ref) | 1.13 (0.58-2.20) | 1.03 (0.52-2.02) | 1.34 (0.69-2.59) | 0.443 |
| Model 2 | 1.00 (ref) | 0.92 (0.45-1.89) | 0.92 (0.45-1.87) | 1.23 (0.61-2.48) | 0.577 |
| Mn | | | | | |
| Range | ≤-0.15 | ~0.74 | ~1.38 | ≥1.39 | |
| Case/Control | 78/36 | 46/36 | 14/36 | 6/36 | |
| Model 1 | 1.00 (ref) | 0.56 (0.31-1.02) | 0.17 (0.08-0.35) | 0.07 (0.03-0.18) | < 0.001 |
| Model 2 | 1.00 (ref) | 0.58 (0.30-1.11) | 0.16 (0.07-0.35) | 0.05 (0.02-0.15) | < 0.001 |
| Fe | | | | | |
| Range | ≤5.11 | ~5.84 | ~6.57 | ≥6.58 | |
| Case/Control | 27/36 | 42/36 | 41/36 | 34/36 | |
| Model 1 | 1.00 (ref) | 1.62 (0.82-3.19) | 1.59 (0.80-3.15) | 1.32 (0.65-2.67) | 0.427 |
| Model 2 | 1.00 (ref) | 1.77 (0.85-3.66) | 1.56 (0.74-3.27) | 1.16 (0.54-2.49) | 0.745 |
| Co | | | | | |
| Range | ≤-1.36 | ~-0.66 | ~-0.02 | ≥-0.01 | |
| Case/Control | 21/36 | 56/36 | 37/36 | 30/36 | |
| Model 1 | 1.00 (ref) | 2.65 (1.34-5.24) | 1.76 (0.85-3.64) | 1.39 (0.65-2.97) | 0.794 |
| Model 2 | 1.00 (ref) | 2.50 (1.21-5.16) | 1.42 (0.65-3.11) | 1.35 (0.60-3.02) | 0.933 |
| Ni | | | | | |
| Range | ≤0.22 | ~0.86 | ~1.52 | ≥1.53 | |
| Case/Control | 19/36 | 50/36 | 51/36 | 24/36 | |
| Model 1 | 1.00 (ref) | 2.61 (1.29-5.28) | 2.68 (1.33-5.40) | 1.23 (0.57-2.66) | 0.698 |
| Model 2 | 1.00 (ref) | 2.90 (1.34-6.30) | 2.93 (1.38-6.25) | 1.51 (0.65-3.52) | 0.458 |

*(Continued)*

**Table 2.** (Continued)

| Elements | Quartile 1 | Quartile 2 | Quartile 3 | Quartile 4 | p trend |
|---|---|---|---|---|---|
| Cu | | | | | |
| Range | ≤0.06 | ~1.97 | ~2.72 | ≥2.73 | |
| Case/Control | 34/36 | 39/36 | 58/36 | 13/36 | |
| Model 1 | 1.00 (ref) | 1.14 (0.59-2.19) | 1.67 (0.89-3.15) | 0.37 (0.17-0.82) | 0.427 |
| Model 2 | 1.00 (ref) | 1.18 (0.58-2.45) | 2.14 (1.04-4.38) | 0.36 (0.15-0.86) | 0.558 |
| Zn | | | | | |
| Range | ≤4.39 | ~5.15 | ~5.91 | ≥5.92 | |
| Case/Control | 25/36 | 50/36 | 48/36 | 21/36 | |
| Model 1 | 1.00 (ref) | 2.06 (1.04-4.06) | 1.97 (0.99-3.91) | 0.85 (0.39-1.79) | 0.707 |
| Model 2 | 1.00 (ref) | 1.92 (0.92-3.99) | 2.45 (1.14-5.26) | 0.76 (0.33-1.74) | 0.700 |
| As | | | | | |
| Range | ≤2.96 | ~3.43 | ~3.82 | ≥3.83 | |
| Case/Control | 35/36 | 35/36 | 35/36 | 39/36 | |
| Model 1 | 1.00 (ref) | 0.99 (0.52-1.93) | 0.99 (0.51-1.92) | 1.10 (0.57-2.12) | 0.787 |
| Model 2 | 1.00 (ref) | 0.99 (0.48-2.01) | 1.13 (0.56-2.30) | 1.09 (0.54-2.19) | 0.745 |
| Rb | | | | | |
| Range | ≤6.43 | ~6.85 | ~7.36 | ~7.37 | |
| Case/Control | 52/36 | 57/36 | 27/36 | 8/36 | |
| Model 1 | 1.00 (ref) | 1.07 (0.58-1.94) | 0.49 (0.26-0.97) | 0.15 (0.06-0.36) | < 0.001 |
| Model 2 | 1.00 (ref) | 1.12 (0.59-2.12) | 0.49 (0.25-1.01) | 0.15 (0.06-0.37) | < 0.001 |
| Sr | | | | | |
| Range | ≤4.11 | ~4.66 | ~5.28 | ~5.29 | |
| Case/Control | 28/36 | 38/36 | 51/36 | 27/36 | |
| Model 1 | 1.00 (ref) | 1..36 (0.69-2.67) | 1.84 (0.96-3.54) | 0.97 (0.48-1.96) | 0.789 |
| Model 2 | 1.00 (ref) | 1.43 (0.69-2.92) | 1.79 (0.89-3.59) | 0.82 (0.38-1.77) | 0.881 |
| Mo | | | | | |
| Range | ≤3.15 | ~3.65 | ~4.32 | ≥4.33 | |
| Case/Control | 26/36 | 48/36 | 51/36 | 19/36 | |
| Model 1 | 1.00 (ref) | 1.87 (0.96-3.65) | 1.99 (1.03-3.87) | 0.69 (0.33-1.48) | 0.456 |
| Model 2 | 1.00 (ref) | 2.10 (1.02-4.31) | 2.20 (1.07-4.51) | 0.73 (0.32-1.66) | 0.555 |
| Te | | | | | |
| Range | ≤-3.78 | ~-3.13 | ~-2.50 | ≥-2.49 | |
| Case/Control | 42/36 | 48/36 | 30/36 | 24/36 | |
| Model 1 | 1.00 (ref) | 1.15 (0.62-2.14) | 0.72 (0.37-1.39) | 0.57 (0.29-1.14) | 0.067 |
| Model 2 | 1.00 (ref) | 1.09 (0.56-2.12) | 0.72 (0.35-1.47) | 0.54 (0.26-1.13) | 0.069 |
| I | | | | | |
| Range | ≤5.07 | ~5.63 | ~6.22 | ≥6.23 | |
| Case/Control | 72/36 | 44/36 | 19/36 | 9/36 | |
| Model 1 | 1.00 (ref) | 0.61 (0.34-1.11) | 0.27 (0.13-0.53) | 0.12 (0.05-0.28) | < 0.001 |
| Model 2 | 1.00 (ref) | 0.63 (0.33-1.18) | 0.22 (0.10-0.47) | 0.10 (0.04-0.25) | < 0.001 |

Model 1: Only adjusted by gender and age (continuous variable).

Model 2: Additionally adjusted for the following covariates: BMI, residence, monthly household income, physical activity, sleep, smoking, alcohol consumption, and family history of mental diseases.

B: boron, Mg: magnesium, Al: aluminum, Ti: titanium, V: vanadium, Mn: manganese, Fe: iron, Co: cobalt, Ni: nickel, Cu: copper, Zn: zinc, As: arsenic, Rb: rubidium, Sr: strontium, Mo: molybdenum, Te: tellurium, I: iodine.

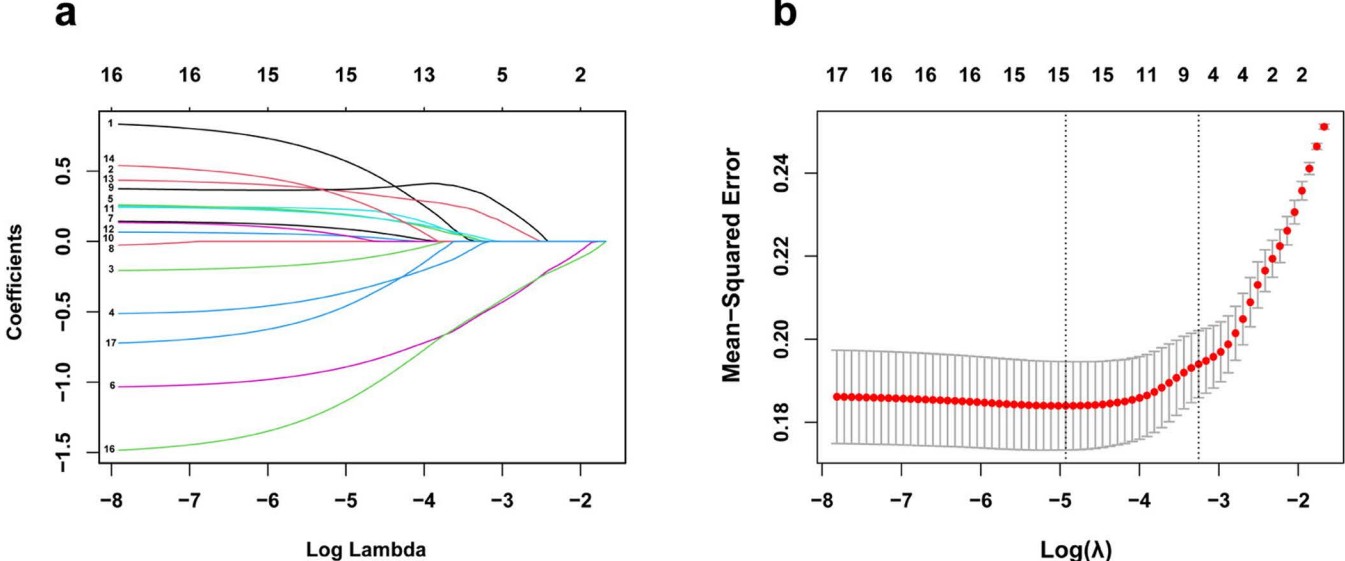

**Fig 1. The elements selected for the multi-element model by LASSO regression.** (a) The change in trajectory of each variable coefficient (λ). (b) Cross-validation plot for the penalty term.

**Table 3. Multi-element model for adolescent BD associated with multiple elements and characteristics of participants.**

| Elements | Quartile 1 | Quartile 2 | Quartile 3 | Quartile 4 | p-trend |
|---|---|---|---|---|---|
| Mg | 1.00 (ref) | 1.79 (0.60-5.30) | 2.91 (0.83-10.24) | 11.24 (1.83-69.12) | 0.013 |
| Ti | 1.00 (ref) | 1.29 (0.53-3.19) | 0.48 (0.17-1.34) | 0.14 (0.04-0.53) | 0.005 |
| Mn | 1.00 (ref) | 0.48 (0.21-1.12) | 0.09 (0.03-0.30) | 0.02 (0.01-0.08) | < 0.001 |
| Ni | 1.00 (ref) | 3.94 (1.43-10.91) | 14.94 (1.49-49.73) | 6.86 (1.55-30.29) | 0.003 |
| Zn | 1.00 (ref) | 2.75 (0.98-7.72) | 3.73 (1.26-11.10) | 2.22 (0.58-8.55) | 0.178 |
| Sr | 1.00 (ref) | 0.80 (0.30-2.18) | 1.29 (0.42-4.04) | 2.18 (0.49-9.76) | 0.521 |
| I | 1.00 (ref) | 0.38 (0.16-0.89) | 0.14 (0.05-0.40) | 0.06 (0.02-0.22) | < 0.001 |

We also analyzed urinary elements during the depressive and manic periods. We identified the representative elements using LASSO regression. In the multi-element model, Mg, Fe, and Sr were positively correlated with the depression phase, whereas Mn, and I showed a negative correlation (Table S5 in S1 File). The BKMR results are consistent with those of the logistic regression analysis (Figs S2 and S3 in S1 File).

In the manic phase, V, Cu, and Sr positively correlated with the manic phase, whereas Mn, and I negatively correlated with the manic phase. According to our analysis, Mn and I were significantly correlated with both the depressive and manic phases (Table S6 in S1 File). Furthermore, Cu interacted significantly with I, Mn, Sr, and V in the BKMR model (Figs S4 and S5 in S1 File).

## Discussion

In this case-control study, we evaluated the association between 17 urinary elements and BD using various statistical models. The findings showed that in the single-element models, urinary Ti, Mn, Rb, and I were negatively associated with adolescent BD. In the multi-element model selected using LASSO, Ti, Mn, and I were negatively correlated with

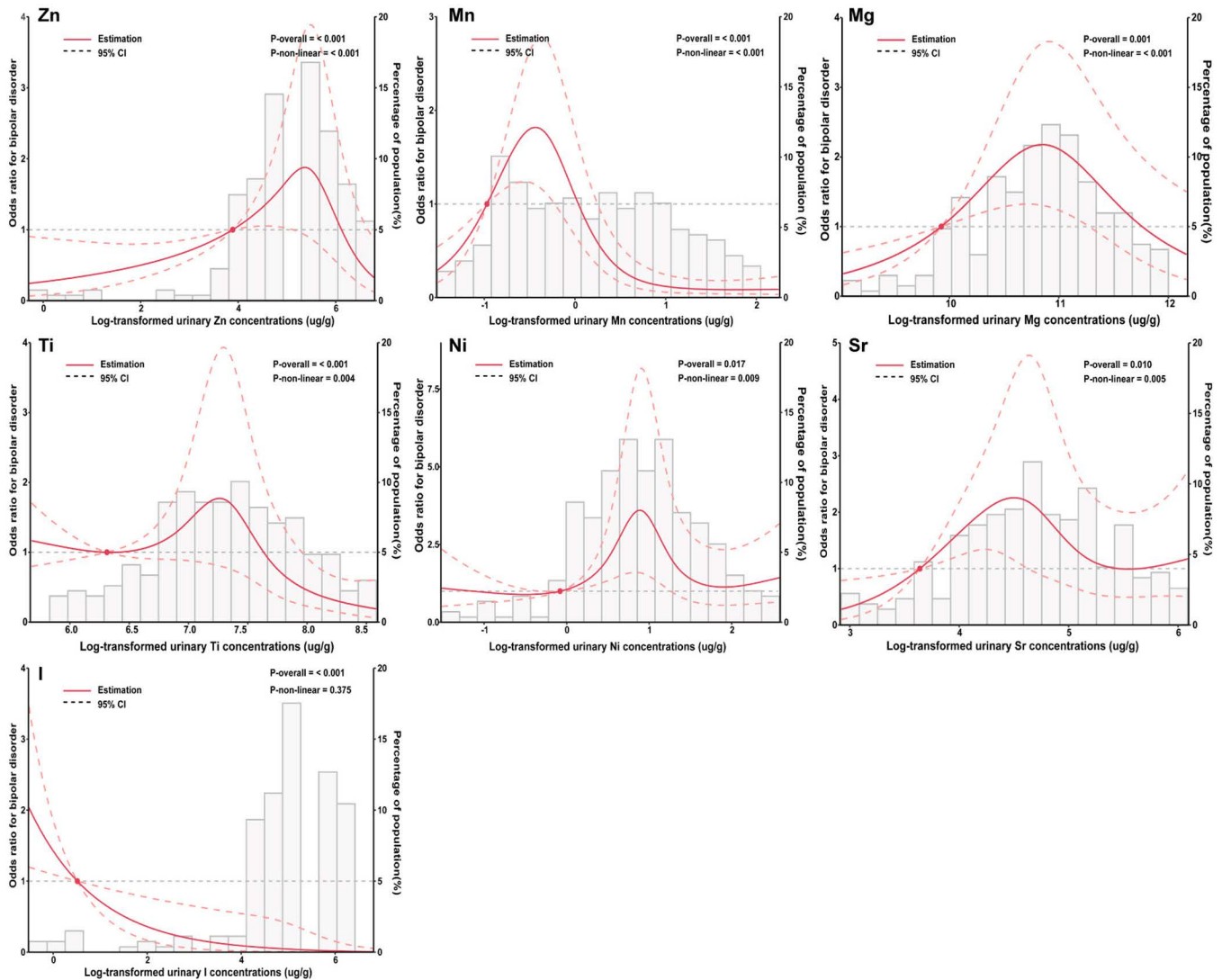

**Fig 2. Adjusted restricted cubic spline (RCS) for the association between urinary elements and bipolar disorder.** The red lines indicate adjusted odds ratios and the dashed lines represent adjusted odds ratios [95% CI] based on RCS for the log-transformed levels of Zn, Mn, Mg, Ti, Ni, Sr and I, with the reference value was set at the 10th percentile. Adjustment factors were gender, age, BMI, residence, household monthly income, physical activity, sleep, smoking, alcohol consumption, family history of mental diseases.

adolescent BD, whereas Mg, Ni, Zn, and Sr were positively correlated. BKMR analyses indicated a significant joint effect of a mixture of seven elements (Ti, Mn, I, Mg, Ni, Zn, and Sr) on BD. The univariate exposure-response function revealed negative relationships between BD and Ti, Mn, and I, while urinary Mg, Ni, Zn, and Sr were positively associated with BD. Furthermore, the bivariate exposure-response function showed that Zn, Mn, and I have a potential correlation with adolescent BD, and that Ni and Mn may also interact.

An interesting finding of our study is that urinary Ti was significantly and negatively associated with adolescent BD. Ti is a common element in nature and is widely used in various products, including in vehicle manufacturing, food colorants, and healthcare industries [28]. Studies have shown titanium dioxide nanoparticle exposure results in microglial activation, reactive oxygen species production, and activation of signaling pathways involved in inflammation and cell death

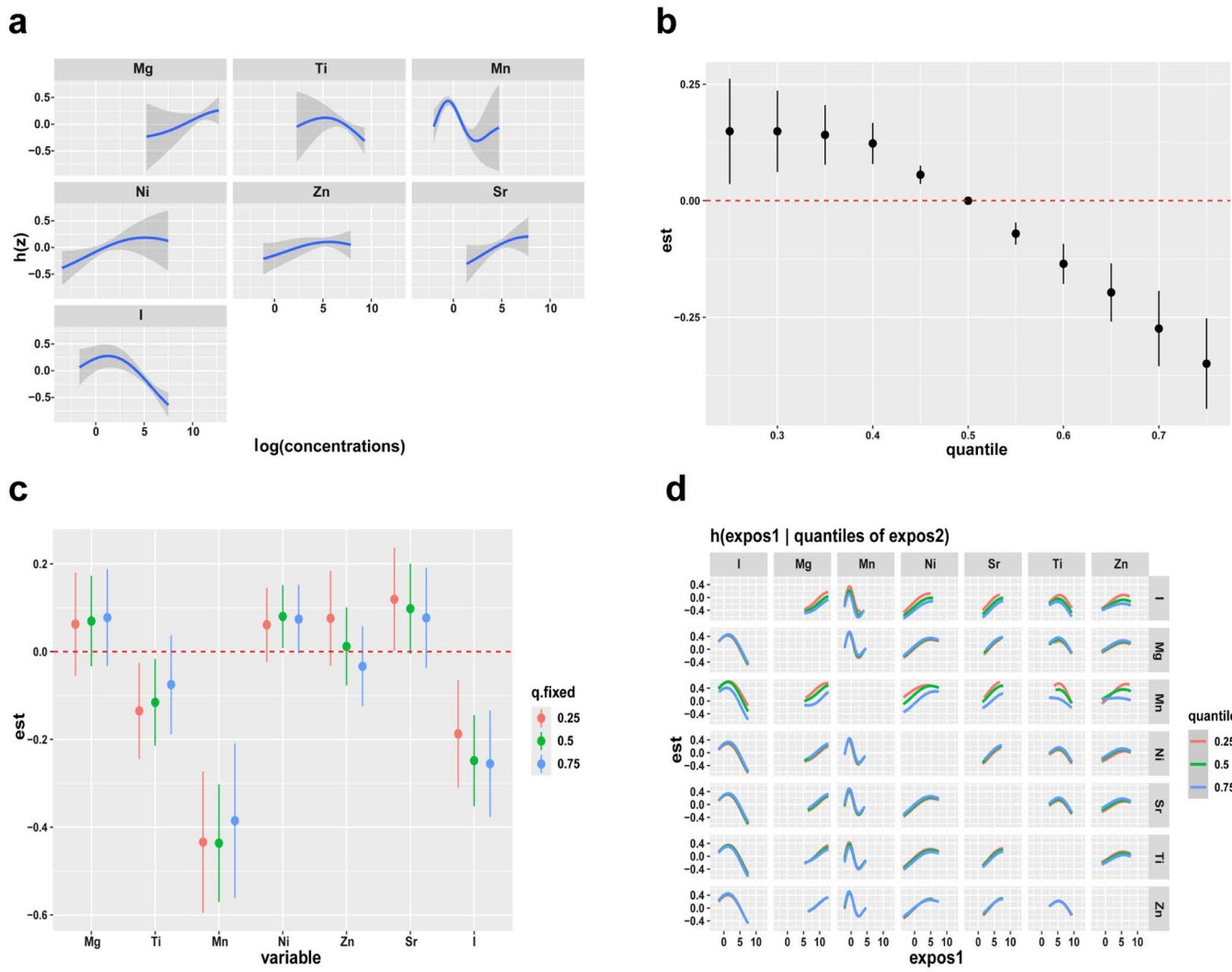

**Fig 3. The BKMR models of each element and mixed exposure effects on BD.** (a) Univariate exposure-response functions of each element (95%CI) with others elements fixed at their medians (P50). (b) The overall effects of mixed exposure, with elements fixed at different percentiles compared to their concentrations when they are at their medians (P50). (c) The effects of single-exposure when an individual element is at its 75th percentile as compared to when that exposure is at its 25th percentile, where all of other exposures are fixed to P75. (d) The bivariate cross-section effects of the exposure-response function of a single element, where the second element is fixed at P25, P50, and P75.

[28]. Abbas et al. reported that exposure to Ti during pregnancy negatively affected hippocampal cell proliferation as well as learning and memory [29]. In a prospective study of a nationally representative sample of children and adolescents, researchers found a significant negative association between urinary Ti levels and intelligence [30]. This is the first study to investigate the effects of Ti on BD in Chinese adolescents. However, the mechanism underlying the interaction between Ti and BD requires further research.

In our study, a significant negative correlation was detected between urinary Mn levels and adolescent BD (p < 0.001). Accumulation of Mn in the brain leads to neurotoxic effects. Neurons in the early developmental stage are especially sensitive to the neurotoxic effects of Mn [31,32]. Animal studies have demonstrated that exposure to excess Mn during the prenatal and postnatal periods leads to tissue Mn deposition in the striatum and hippocampus, which are important brain

regions for cognitive function [33,34]. Some studies have revealed a negative relationship between Mn exposure and children's intelligence [35]. A birth cohort study in China reported that urinary Mn concentrations were positively associated with the Performance IQ of school-age children, particularly girls [36].

Moreover, there was a negative association between I and BD in adolescents. Iodine, is an essential micronutrient necessary for the synthesis of thyroid hormones (TH) [37]. Lack of TH may impair normal neurogenesis and reduces the number of newborn neuroblasts and immature neurons, leading to psychological disorders [38]. A study on children and adolescents found that high urinary I concentration was strongly correlated with a decreased risk of depression [39]. Interestingly, in adult studies, high urinary I levels were associated with a higher prevalence of depressive symptoms [40,41].

In our study, urinary Mg and Ni levels were positively associated with BD in adolescents. These results are similar to those of a previous study that found a positive association between Mg and adult BD [10]. Magnesium is a crucial micronutrients in brain function, mood regulation, and neuronal health [42]. Existing research indicates that the limbic-hypothalamic-pituitary-adrenocortical axis is more sensitive to Mg than other elements [43], and it can suppress hippocampal kindling, reduce the release of adrenocorticotrophic hormone (ACTH), and affect adrenocortical sensitivity to ACTH [44]. Ni primarily stimulates neurotoxicity by inducing oxidative stress, which leads to impairments in neuronal function and altered neurotransmission, generating reactive oxygen species and oxidative stress stimuli, which are involved in the onset and progression of neuropsychiatric disorders [45]. A case-control study suggested that lower Ni level are associated with an increased risk of major depressive disorder (MDD) [46].

Our study found no statistically significant associations between Zn and Sr levels and BD in adolescents. However, the univariate exposure–response function revealed that Zn and Sr were positively associated with BD. Low Zn levels impair the body's ability to resist oxidation, resulting in oxidative damage caused by free radicals [47]. Research has found that patients with MDD have lower serum Zn levels than controls, indicating that lower Zn levels are correlated with an increased risk of MDD [48]. A study conducted in China reported a positive association between Sr concentrations and MDD risk in older Chinese women [49]. Large-scale clinical trials and longitudinal studies are required to investigate causality between Zn, Sr, and BD.

Individuals with BD often experience chronic neuroinflammation and disruptions in neurotransmitter systems such as dopamine and serotonin. The synthesis and metabolism of these neurotransmitters rely on multiple trace elements [50]. For example, trace elements such as Zn and Mn are essential for the enzymatic reactions involved in dopamine and serotonin synthesis [51]. Episodes of mania or depression can trigger oxidative stress and metabolic abnormalities, which may impair the absorption and utilization of trace elements [52]. In addition, the gut microbiome of patients with BD differs from that of healthy individuals, showing reduced microbial diversity and altered microbial abundance. This disruption can affect the absorption and utilization of trace elements, contributing to changes in their levels [53,54]. In conclusion, the causes of elevated trace element levels in patients with BD may be multifactorial and include altered neurotransmitter function, oxidative stress, and changes in gut microbiota composition.

Although pharmacological treatment can influence trace element levels, either by increasing or decreasing specific elements, it is not considered a primary factor in the observed changes in urinary trace element concentrations. For example, antipsychotic medications may partially explain fluctuations in trace element levels (e.g., Cu and Fe), either by increasing or decreasing them [55,56]. However, other study found that the levels of elements (Mn, Mg, Zn, etc) in bipolar patients after treatment did not differ significantly from those in healthy individuals [57,58]. Moreover, while low dietary intakes of Mn and Zn have been associated with symptoms of depression and anxiety, Mg intake did not show a significant relationship [59]. A nationwide population-based study found no significant association between Li levels in drinking water and BD prevalence [60]. Therefore, pharmacological treatment and dietary factors may influence trace element concentrations, however, these factors are not considered the primary causes of the observed changes in urinary trace element levels. In addition, we have summarized ongoing and completed clinical trials, as well as existing clinical and basic research relevant to our study (Tables S7, S8, and S9 in S1 File).

Our study has several advantages. First, we focused on adolescent BD and attempted to explain its etiology in Chinese adolescents. In addition to studying the effects of single elements, we utilized the BKMR model to estimate the relative contributions of individual elements and investigated the effects of multiple elements and mixed exposures on BD. Nevertheless, our study has several limitations. First, we did not specifically assess the potential influence of pharmacological treatments or lifestyle factors (diet, exercise, and smoking) on urinary trace element concentrations. These factors may affect trace element levels. Future research should investigate these influences more directly to provide a clearer understanding of their role in trace element alterations in BD. Second, although this case-control study identified potential associations between the disease and exposure factors, it failed to establish a definitive causal link. Additionally, blood samples can be difficult to obtain from children, so urine samples are valuable for biological detection. The concentrations of elements in urine can be influenced by urine dilution, which often means that the results may not accurately reflect the true levels. Finally, because all the adolescents included in the study were Chinese, the findings may not be fully applicable or generalizable to adolescent populations in other countries.

## Conclusions

In conclusion, this study indicated that there are differences in trace elements between adolescent patients with BD and healthy controls. Specifically, Ti, Mn, and I levels negatively correlated with BD in adolescents, while Mg and Ni levels positively correlated with BD. Furthermore, the combined effect of seven elements (Ti, Mn, I, Mg, Ni, Zn, and Sr) was significantly correlated with BD. These findings extend our understanding of the relationships between multiple trace elements and adolescent BD.

## Supporting information

**S1 File. Supplementary material.**
(DOCX)

## Author contributions

**Conceptualization:** Jie Li, Xuemei Li, Yuqian He.

**Data curation:** Jie Li, Yuqian He, Yajie Huang, Wenjing Wang, Chengzhi Chen, Hang Du.

**Formal analysis:** Jie Li, Xuemei Li.

**Investigation:** Xinyu Zhou, Jie Li, Yuqian He, Yajie Huang, Wenjing Wang, Chengzhi Chen, Hang Du, Dan Zhu.

**Methodology:** Jie Li, Xuemei Li, Yuqian He.

**Project administration:** Xinyu Zhou, Dan Zhu.

**Software:** Jie Li, Xuemei Li, Yuqian He.

**Supervision:** Xinyu Zhou, Dan Zhu.

**Validation:** Jie Li.

**Writing – original draft:** Jie Li, Xuemei Li.

**Writing – review & editing:** Xinyu Zhou, Jie Li, Xuemei Li, Yuqian He, Yajie Huang, Wenjing Wang, Dan Zhu.

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
