## [Decision Letter · Decision Letter 0]

22 Dec 2024

PONE-D-24-46821Associations of Multiple Trace Elements with the Risk of Bipolar Disorder in Adolescents: A Case-Control StudyPLOS ONE

Dear Dr. Xinyu,

Thank you for submitting your manuscript to PLOS ONE. After careful consideration, we feel that it has merit but does not fully meet PLOS ONE’s publication criteria as it currently stands. Therefore, we invite you to submit a revised version of the manuscript that addresses the points raised during the review process.

We look forward to receiving your revised manuscript.

Kind regards,

Li Yang, M.D.

Academic Editor

PLOS ONE

Journal Requirements:

2. In this instance it seems there may be acceptable restrictions in place that prevent the public sharing of your minimal data. However, in line with our goal of ensuring long-term data availability to all interested researchers, PLOS’ Data Policy states that authors cannot be the sole named individuals responsible for ensuring data access (http://journals.plos.org/plosone/s/data-availability#loc-acceptable-data-sharing-methods ).

Reviewers' comments:

Reviewer's Responses to Questions

**Comments to the Author**

1. Is the manuscript technically sound, and do the data support the conclusions?

Reviewer #1: Yes

Reviewer #2: Partly

Reviewer #3: Yes

2. Has the statistical analysis been performed appropriately and rigorously? 

Reviewer #1: I Don't Know

Reviewer #2: Yes

Reviewer #3: Yes

3. Have the authors made all data underlying the findings in their manuscript fully available?

Reviewer #1: No

Reviewer #2: Yes

Reviewer #3: Yes

4. Is the manuscript presented in an intelligible fashion and written in standard English?

Reviewer #1: Yes

Reviewer #2: No

Reviewer #3: Yes

5. Review Comments to the Author

Reviewer #1: Dear Authors,

Your manuscript is highly interesting, as it provides valuable insights into elements that contribute to the diagnosis, treatment, and overall management of psychiatric conditions. Expanding our knowledge in this area is essential.

Despite this, some amendments are necessary to improve its quality.

1. Did the BD participants receive pharmacological treatment? Could this treatment have influenced the urinary elements?

2. Into the results, rewrite the sentences such as: "Significant differences were observed between the cases and controls in terms of physical activity, suicidal behavior, NSSI, YMRS, HAMD-17, and HAMA-14". It is not clear what are the differences between the two groups. Despite there is the table 1, it is preferable to write the results also in narrative form.

Reviewer #2: I have read the manuscript carefully, and my comments are as below:

1. The title of the article "Associations of Multiple Trace Elements with the Risk of Bipolar Disorder in

Adolescents: A Case-Control Study" is misleading, and it should be revised.

2. How do you define "the risk of Biopolar Disroder"? This variable is not clearly defined in the manuscript.

3. In this work, 144 clinically defined BD subjects were selected along with the matching number of control normal subjects. The urine was analyzed for trace elements, and the statistical analysis was carried out.

In plain words, there is a correlation between urinary elemental concentration among the BD subjects compared to normal subjects.

4. Authors should comment on the possible reasons for elevated trace elements. Is it due to medication given to the BD subjects or due to food/water intake? The authors need to find out why there are differences in elemental concentrations.

5. Similarly conclusions: "We observed inhibitory effect of titanium, manganese, and iodine on BD

risk, on the contrary, magnesium, and nickel may increase the risk of BD." are not supported by any data in the work. How are Mg and Ni responsible for increasing the risk of BD, and how have the authors arrived at this conclusion just based on the analytical studies?

6. The claim that "These findings will provide valuable insights into the potential influence of trace elements on

the pathophysiology of BD and may have significant implications for prevention and treatment strategies. " is completely far fetched. There is no supporting evidence for these statements in the work carried out by authors.

Hence, I suggest that the authors should rewrite the manuscript and qualify their claims based on the findings.

Reviewer #3: This case-control study investigates the association between various trace elements and the risk of bipolar disorder in adolescents.

The introduction clearly states the study's initial hypotheses and outlines its objectives. It also explains the statistical methods used to identify any potential relationships between urinary levels of multiple trace elements and bipolar disorder. Additionally, this section could benefit from further elaboration on the links between key elements studied and bipolar disorder, based on existing evidence. This would enhance readers' understanding and provide a better context for interpreting the study's results.

In the methods section, the study details the population under investigation, the timeframe and location of the research, the criteria for inclusion and exclusion, the questionnaires used, and the techniques employed to analyze the results.

The discussion section effectively compares the study's findings with those of similar research and is well-structured overall.

6. PLOS authors have the option to publish the peer review history of their article (what does this mean? ). If published, this will include your full peer review and any attached files.

**Do you want your identity to be public for this peer review?** For information about this choice, including consent withdrawal, please see our Privacy Policy .

Reviewer #1: **Yes: ** Francesco Maria Boccaccio

Reviewer #2: No

Reviewer #3: **Yes: ** Rahim Badrfam

---

## [Author Response · Author response to Decision Letter 0]

4 Feb 2025

REVIEWER' GENERAL POINTS:

Reviewer #1: Your manuscript is highly interesting, as it provides valuable insights into elements that contribute to the diagnosis, treatment, and overall management of psychiatric conditions. Expanding our knowledge in this area is essential.

Authors’ response:

Thank you very much for your positive and helpful comments on our efforts. We have carefully revised the paper according to your suggestions point by point.

Major points 1: Did the BD participants receive pharmacological treatment? Could this treatment have influenced the urinary elements?

Authors’ response:

Thank you for your valuable comments. In our study, 139 patients with bipolar disorder were receiving pharmacological treatment, and we added the relevant content in Table 1. While pharmacological treatment can influence trace element levels - either increasing or decreasing specific elements - it is not considered the primary factor the observed changes in urinary trace element concentrations. We have expanded the discussion to reflect this, with the following addition: "While pharmacological treatment can influence trace element levels - either increasing or decreasing specific elements - it is not considered the primary factor the observed changes in urinary trace element concentrations. For example, antipsychotic medications may partially explain fluctuations in trace element levels (Cu, Fe, etc.), either increasing or decreasing them (Wolf et al. 2006, Kabzińska-Milewska et al. 2021). However, other study found that the levels of elements (Mn, Mg, Zn, etc) in bipolar patients after treatment did not differ significantly from those in healthy individuals(Sampath et al. 2022, Rog et al. 2024)." (see line 412 in the revised manuscript).

We also acknowledge the following limitations in our study: "Firstly, our study did not specifically assess the potential influence of pharmacological treatment or lifestyle factors (such as diet, exercise, and smoking) on urinary trace element concentrations. These factors may affect trace element levels. Future research should aim to investigate these influences more directly, providing a clearer understanding of their role in trace element alterations in BD." (see line 428 in the revised manuscript).

Reference

Wolf, T. L., J. Kotun and J. H. Meador-Woodruff (2006). "Plasma copper, iron, ceruloplasmin and ferroxidase activity in schizophrenia." Schizophr Res 86(1-3): 167-171.

Kabzińska-Milewska, K., R. W. Wójciak and D. J. J. o. E. Czajeczny (2021). "Effect of lithium treatment on the content of lithium, copper, calcium, magnesium, zinc and iron in the hair of patients with bipolar disorder."

Rog, J., L. Lobejko, M. Hordejuk, W. Marciniak, R. Derkacz, A. Kiljanczyk, M. Matuszczak, J. Lubinski, M. Nesterowicz, M. Zendzian-Piotrowska, A. Zalewska, M. Maciejczyk and H. Karakula-Juchnowicz (2024). "Pro/antioxidant status and selenium, zinc and arsenic concentration in patients with bipolar disorder treated with lithium and valproic acid." Front Mol Neurosci 17: 1441575.

Sampath, V. P., S. V. Singh, I. Pelov, O. Tirosh, Y. Erel and D. Lichtstein (2022). "Chemical Element Profiling in the Sera and Brain of Bipolar Disorders Patients and Healthy Controls." Int J Mol Sci 23(22).

Major points 2: Into the results, rewrite the sentences such as: "Significant differences were observed between the cases and controls in terms of physical activity, suicidal behavior, NSSI, YMRS, HAMD-17, and HAMA-14". It is not clear what are the differences between the two groups. Despite there is the table 1, it is preferable to write the results also in narrative form.

Authors’ response:

Thank you for your suggestion. We have revised the sentence to provide clearer. The revised version reads as follows: “Significant differences were observed between the cases and control groups in physical activity (p = 0.009) and suicidal behavior (p < 0.001). In the case group, 67.4% of patients exhibited non-suicidal self-injury (NSSI) behavior, with the majority experiencing current mood depression (61.8%). Additionally, 96.6% of these patients were receiving treatment. Scores on the YMRS (p < 0.001), HAMD-17 (p < 0.001), and HAMA-14 (p < 0.001) were significantly higher in the case group compared to the control group." (see line 213 in the revised manuscript).

Reviewer #2: I have read the manuscript carefully, and my comments are as below:

Authors’ response:

Thanks very much for your careful reviewing of this manuscript and the constructive suggestion. We have carefully revised the paper according to your suggestions.

Major points 1 and 2: The title of the article "Associations of Multiple Trace Elements with the Risk of Bipolar Disorder in Adolescents: A Case-Control Study" is misleading, and it should be revised. How do you define "the risk of Bipolar Disorder"? This variable is not clearly defined in the manuscript.

Authors’ response:

Thank you for your suggestion. We agree with you, and we have revised the title to: "Associations of Multiple Trace Elements with Bipolar Disorder in Adolescents: A Case-Control Study." Meanwhile, we have thoroughly revised the manuscript to address any ambiguities, ensuring that all terms and variables are clearly defined.

Major points 3 and 4: In this work, 144 clinically defined BD subjects were selected along with the matching number of control normal subjects. The urine was analyzed for trace elements, and the statistical analysis was carried out. In plain words, there is a correlation between urinary elemental concentration among the BD subjects compared to normal subjects. Authors should comment on the possible reasons for elevated trace elements. Is it due to medication given to the BD subjects or due to food/water intake? The authors need to find out why there are differences in elemental concentrations.

Authors’ response:

Thank you for your insightful comments. According to your suggestion, we have expanded the discussion to address the potential causes of elevated trace elements in BD patients. In summary, the changes in urinary trace element levels in BD are likely influenced by a combination of neuroinflammation, neurotransmitter disturbances, oxidative stress, medication effects, altered gut microbiota, and dietary or environmental factors.

We have incorporated the following addition to reflect these factors more comprehensively: "Individuals with BD often experience chronic neuroinflammation and disruptions in neurotransmitters systems such as dopamine and serotonin. The synthesis and metabolism of these neurotransmitters rely on multiple trace elements (Anderson et al. 2012). For example, trace elements like Zn and Mn are essential for the enzymatic reactions involved in dopamine and serotonin synthesis (Gao et al. 2024). Episodes of mania or depression can trigger oxidative stress and metabolic abnormalities, which may impair the absorption and utilization of trace elements (Shazia et al. 2012). Additionally, the gut microbiome in BD patients differs from that of healthy individuals, showing reduced microbial diversity and altered microbial abundance. This disruption can affect the absorption and utilization of trace elements, contributing to changes in their levels (Coello et al. 2019, Ortega et al. 2023). In conclusion, the causes of elevated trace elements in BD patients may be multifactorial, including altered neurotransmitter function, oxidative stress, and changes in gut microbiota composition." (see line 401 in the revised manuscript).

Additionally, we discussed the influence of pharmacological treatments:“ While pharmacological treatment can influence trace element levels - either increasing or decreasing specific elements - it is not considered the primary factor the observed changes in urinary trace element concentrations. For example, antipsychotic medications may partially explain fluctuations in trace element levels (Cu, Fe, etc.), either increasing or decreasing them (Wolf et al. 2006, Kabzińska-Milewska et al. 2021). However, other study found that the levels of elements (Mn, Mg, Zn, etc) in bipolar patients after treatment did not differ significantly from those in healthy individuals (Sampath et al. 2022, Rog et al. 2024). Moreover, while low dietary intake of Mn and Zn has been associated with symptoms of depression and anxiety, Mg intake did not show a significant relationship (Nakamura et al. 2019). A nationwide population-based study found no statistical significance between the lithium levels in drinking water and BD prevalence (Kessing et al. 2017). Therefore pharmacological treatment and dietary factors may influence trace element concentrations, these factors are not considered the primary causes of the observed changes in urinary trace element levels." (see line 412 in the revised manuscript).

Finally, We also acknowledge the following limitations in our study: "Firstly, our study did not specifically assess the potential influence of pharmacological treatment or lifestyle factors (such as diet, exercise, and smoking) on urinary trace element concentrations. These factors may affect trace element levels. Future research should aim to investigate these influences more directly, providing a clearer understanding of their role in trace element alterations in BD." (see line 428 in the revised manuscript).

Reference

Anderson, I. M., P. M. Haddad and J. Scott (2012). "Bipolar disorder." BMJ 345: e8508.

Coello, K., T. H. Hansen, N. Sorensen, K. Munkholm, L. V. Kessing, O. Pedersen and M. Vinberg (2019). "Gut microbiota composition in patients with newly diagnosed bipolar disorder and their unaffected first-degree relatives." Brain Behav Immun 75: 112-118.

Gao, B., C. Li, Y. Qu, M. Cai, Q. Zhou, Y. Zhang, H. Lu, Y. Tang, H. Li and H. Shen (2024). "Progress and trends of research on mineral elements for depression." Heliyon 10(15): e35469.

Kabzińska-Milewska, K., R. W. Wójciak and D. J. J. o. E. Czajeczny (2021). "Effect of lithium treatment on the content of lithium, copper, calcium, magnesium, zinc and iron in the hair of patients with bipolar disorder."

Kessing, L. V., T. A. Gerds, N. N. Knudsen, L. F. Jorgensen, S. M. Kristiansen, D. Voutchkova, V. Ernstsen, J. Schullehner, B. Hansen, P. K. Andersen and A. K. Ersboll (2017). "Lithium in drinking water and the incidence of bipolar disorder: A nation-wide population-based study." Bipolar Disord 19(7): 563-567.

Nakamura, M., A. Miura, T. Nagahata, Y. Shibata, E. Okada and T. Ojima (2019). "Low Zinc, Copper, and Manganese Intake is Associated with Depression and Anxiety Symptoms in the Japanese Working Population: Findings from the Eating Habit and Well-Being Study." Nutrients 11(4).

Ortega, M. A., M. A. Alvarez-Mon, C. Garcia-Montero, O. Fraile-Martinez, J. Monserrat, L. Martinez-Rozas, R. Rodriguez-Jimenez, M. Alvarez-Mon and G. Lahera (2023). "Microbiota-gut-brain axis mechanisms in the complex network of bipolar disorders: potential clinical implications and translational opportunities." Mol Psychiatry 28(7): 2645-2673.

Rog, J., L. Lobejko, M. Hordejuk, W. Marciniak, R. Derkacz, A. Kiljanczyk, M. Matuszczak, J. Lubinski, M. Nesterowicz, M. Zendzian-Piotrowska, A. Zalewska, M. Maciejczyk and H. Karakula-Juchnowicz (2024). "Pro/antioxidant status and selenium, zinc and arsenic concentration in patients with bipolar disorder treated with lithium and valproic acid." Front Mol Neurosci 17: 1441575.

Sampath, V. P., S. V. Singh, I. Pelov, O. Tirosh, Y. Erel and D. Lichtstein (2022). "Chemical Element Profiling in the Sera and Brain of Bipolar Disorders Patients and Healthy Controls." Int J Mol Sci 23(22).

Shazia, Q., Z. H. Mohammad, T. Rahman and H. U. Shekhar (2012). "Correlation of oxidative stress with serum trace element levels and antioxidant enzyme status in Beta thalassemia major patients: a review of the literature." Anemia 2012: 270923.

Wolf, T. L., J. Kotun and J. H. Meador-Woodruff (2006). "Plasma copper, iron, ceruloplasmin and ferroxidase activity in schizophrenia." Schizophr Res 86(1-3): 167-171.

Major points 5 and 6: Similarly conclusions: "We observed inhibitory effect of titanium, manganese, and iodine on BD risk, on the contrary, magnesium, and nickel may increase the risk of BD." are not supported by any data in the work. How are Mg and Ni responsible for increasing the risk of BD, and how have the authors arrived at this conclusion just based on the analytical studies? The claim that "These findings will provide valuable insights into the potential influence of trace elements on the pathophysiology of BD and may have significant implications for prevention and treatment strategies. " is completely far fetched. There is no supporting evidence for these statements in the work carried out by authors.

Authors’ response:

Thanks for your suggestion. According to your advice, we have revised the abstract in the manuscript as outlined below: "RCS results showed a non-linear correlation between the elements titanium, manganese, iodine, magnesium, nickel, zinc, strontium and adolescent BD. In addition, BKMR analysis showed a significant joint effect of multiple elements on adolescent BD when the concentrations of seven elements were at or above their 55th percentile compared with their median values." (see line 60 in the revised manuscript)

"Our findings revealed that urinary titanium, manganese and iodine were negatively correlated with adolescent BD, while urinary magnesium and nickel were positively correlated with adolescent BD. These results provide some evidence of an association between urine trace elements and adolescent BD." (see line 64 in the revised manuscript)

In the results section, we have modified as below: "We have observed the potential joint effect of Zn, Mn and I, and the interaction between Ni and Mn." (see line 319 in the revised manuscript)

In the discussion section, we have modified as below: "BKMR analyses indicated a significant joint effect of mixture of seven elements (Ti, Mn, I, Mg, Ni, Zn, and Sr) on BD. The univariate exposure-response function revealed negative relationships between Ti, Mn, and I with BD, while urinary Mg, Ni, Zn, and Sr was positively associated with BD. Furthermore, the bivariate expose-response function shows that Zn, Mn and I have a potential correlation with adolescent BD. and Ni and Mn may also interact." (see line 350 in the revised manuscript)

In the conclusion section, we have modified as follows: "Specifically, levels of Ti, Mn, and I were negatively correlated with BD in adolescents, while levels of Mg and Ni were positively correlated with BD. Furthermore, the combined effects of these seven elements (Ti, Mn, I, Mg, Ni, Zn and Sr) exhibited a significant correlation with BD." (see line 442 in the revised manuscript)

Reviewer #3: This case-control study investigates the association between various trace elements and the risk of bipolar disorder in adolescents.

The introduction clearly states the study's initial hypotheses and outlines its objectives. It also explains the statistical methods used to identify any potential relationships between urinary levels of multiple trace elements and bipolar disorder. Additionally, this section could benefit from further elaboration on the links between key elements studied and bipolar disorder, based on existing evidence. This would enhance readers' understanding and provide a better context for interpreting the study's results.

In the methods section, the study details the population under investigation, the timeframe and location of the research, the criteria for inclusion and exclusion, the questionnaires used, and the techniques employed to analyze the results.

The discussion section effectively compares the study's findings with those of similar research and is well-structured overall.

Authors’ response:

Thank you very much for your thoughtful and detailed review of our manuscript. We greatly appreciate your positive feedback on the study's introduction, methods, and discussion sections.

Once again, thank you for your comments and suggestions. We tried our best to improve the manuscript and made some changes marked in red in revised paper which will not influence the content and framework of the paper. We appreciate for Editors/Reviewers warm wor

---

## [Decision Letter · Decision Letter 1]

7 Mar 2025

PONE-D-24-46821R1Associations of Multiple Trace Elements with Bipolar Disorder in Adolescents: A Case-Control StudyPLOS ONE

Dear Dr. Xinyu,

Thank you for submitting your manuscript to PLOS ONE. After careful consideration, we feel that it has merit but does not fully meet PLOS ONE’s publication criteria as it currently stands. Therefore, we invite you to submit a revised version of the manuscript that addresses the points raised during the review process.

Editor Comments:Thanks for submitting your revised paper to PLOS ONE. Your manuscript has now been assessed by our editorial team and previous peer experts, and I am pleased to inform you that your revised work has been approved by the reviewers. However, before I can recommend the final editorial decision to our journal office, some minor issues need your attention.

1) Please consider to add a table to comprehensively summarize the ongoing/completed clinical trials or existing clinical/basic studies that are related to your study topic.

2) Polish the English language.

We look forward to receiving your revised manuscript.

Kind regards,

Li Yang, M.D.

Academic Editor

PLOS ONE

Journal Requirements:

Reviewers' comments:

Reviewer's Responses to Questions

**Comments to the Author**

1. If the authors have adequately addressed your comments raised in a previous round of review and you feel that this manuscript is now acceptable for publication, you may indicate that here to bypass the “Comments to the Author” section, enter your conflict of interest statement in the “Confidential to Editor” section, and submit your "Accept" recommendation.

Reviewer #2: All comments have been addressed

Reviewer #3: All comments have been addressed

2. Is the manuscript technically sound, and do the data support the conclusions?

Reviewer #2: Yes

Reviewer #3: Yes

3. Has the statistical analysis been performed appropriately and rigorously? 

Reviewer #2: Yes

Reviewer #3: Yes

4. Have the authors made all data underlying the findings in their manuscript fully available?

Reviewer #2: Yes

Reviewer #3: Yes

5. Is the manuscript presented in an intelligible fashion and written in standard English?

Reviewer #2: Yes

Reviewer #3: Yes

6. Review Comments to the Author

Reviewer #2: (No Response)

Reviewer #3: I hope the results of this study will benefit patients. Thank you!

The responses to questions and ambiguities led to improved manuscript quality.

The additional material, particularly in the discussion section, effectively describes the research objectives of the study, notably for patients with bipolar disorder.

7. PLOS authors have the option to publish the peer review history of their article (what does this mean? ). If published, this will include your full peer review and any attached files.

**Do you want your identity to be public for this peer review?** For information about this choice, including consent withdrawal, please see our Privacy Policy .

Reviewer #2: No

Reviewer #3: **Yes: ** Rahim Badrfam

---

## [Author Response · Author response to Decision Letter 1]

30 Mar 2025

EDITOR'S GENERAL POINTS:

Major points 1: Please consider to add a table to comprehensively summarize the ongoing/completed clinical trials or existing clinical/basic studies that are related to your study topic.

Authors’ response:

Thank you for your suggestion. We have added tables summarizing relevant ongoing and completed clinical trials, as well as existing clinical and basic studies related to our research topic. Meanwhile, in the discussion section, we have added it as follow: “In addition, we have summarized ongoing and completed clinical trials, as well as existing clinical and basic research relevant to our study (Tables S7, S8, and S9).” (see line 427 in the revised manuscript)

Major points 2: Polish the English language.

Authors’ response:

Thanks for your suggestion. We have conducted a comprehensive review of the manuscript to improve sentence structure, grammar, and vocabulary. Meanwhile, we have also carried out professional retouching, and multiple rounds of proofreading ensured clarity and fluency.

---

## [Editor Report · Decision Letter 2]

31 Mar 2025

Associations of Multiple Trace Elements with Bipolar Disorder in Adolescents: A Case-Control Study

PONE-D-24-46821R2

Dear Dr. Xinyu,

We’re pleased to inform you that your manuscript has been judged scientifically suitable for publication and will be formally accepted for publication once it meets all outstanding technical requirements.

Kind regards,

Li Yang, M.D.

Academic Editor

PLOS ONE

Additional Editor Comments (optional):

Thanks for the authors' efforts to comprehensively improve your manuscript according to editor's and reviewers' comments. I am pleased to inform you that your paper can be accepted for publication now. Thanks for the chance to assess your interesting and important work. Additionally, many thanks for all the reviewers' precious inputs.
---

## [Editor Report · Acceptance letter]

PONE-D-24-46821R2

PLOS ONE

Dear Dr. Zhou,

I'm pleased to inform you that your manuscript has been deemed suitable for publication in PLOS ONE. Congratulations! Your manuscript is now being handed over to our production team.

Kind regards,

on behalf of

Dr. Li Yang

Academic Editor

PLOS ONE